# Radiomic Signatures Associated with CD8^+^ Tumour-Infiltrating Lymphocytes: A Systematic Review and Quality Assessment Study

**DOI:** 10.3390/cancers14153656

**Published:** 2022-07-27

**Authors:** Syafiq Ramlee, David Hulse, Kinga Bernatowicz, Raquel Pérez-López, Evis Sala, Luigi Aloj

**Affiliations:** 1Department of Radiology, University of Cambridge, Cambridge CB2 0QQ, UK; dgh42@cam.ac.uk (D.H.); es220@medschl.cam.ac.uk (E.S.); la398@medschl.cam.ac.uk (L.A.); 2Radiomics Group, Vall d’Hebron Institute of Oncology (VHIO), 08035 Barcelona, Spain; kbernatowicz@vhio.net (K.B.); rperez@vhio.net (R.P.-L.); 3Department of Radiology, Vall d’Hebron University Hospital, 08035 Barcelona, Spain

**Keywords:** radiomics, cancer, systematic review, immunotherapy, immune cells, lymphocytes

## Abstract

**Simple Summary:**

Immune checkpoint inhibitors can be effective drugs to treat cancer. However, only a minority of patients derive benefits. An important determinant of treatment success is the abundance of CD8-expressing tumour-infiltrating lymphocytes (CD8^+^ TILs) in target tumours. The measurement of CD8^+^ TIL density in the clinical setting relies on tissue sampling. Radiomics, the process of extracting a large number of features from radiological images, may offer a non-invasive alternative. The premise of radiomics is that features on medical images are linked to the underlying molecular, physiological, and structural properties of the tumour. In this systematic review, we address available evidence linking imaging features of tumours with levels of CD8^+^ TILs.

**Abstract:**

The tumour immune microenvironment influences the efficacy of immune checkpoint inhibitors. Within this microenvironment are CD8-expressing tumour-infiltrating lymphocytes (CD8^+^ TILs), which are an important mediator and marker of anti-tumour response. In practice, the assessment of CD8^+^ TILs via tissue sampling involves logistical challenges. Radiomics, the high-throughput extraction of features from medical images, may offer a novel and non-invasive alternative. We performed a systematic review of the available literature reporting radiomic signatures associated with CD8^+^ TILs. We also aimed to evaluate the methodological quality of the identified studies using the Radiomics Quality Score (RQS) tool, and the risk of bias and applicability with the Quality Assessment of Diagnostic Accuracy Studies (QUADAS-2) tool. Articles were searched from inception until 31 December 2021, in three electronic databases, and screened against eligibility criteria. Twenty-seven articles were included. A wide variety of cancers have been studied. The reported radiomic signatures were heterogeneous, with very limited reproducibility between studies of the same cancer group. The overall quality of studies was found to be less than desirable (mean RQS = 33.3%), indicating a need for technical maturation. Some potential avenues for further investigation are also discussed.

## 1. Introduction

Immune checkpoint inhibitors (ICIs) leverage the high specificity of the immune system to selectively attack tumour cells. They can be highly effective [1]. However, only a small proportion of patients meaningfully benefit from treatment, with typical response rates of under 15% for most eligible cancer types [2,3,4]. Therefore, markers that can distinguish between responsive and non-responsive patients are necessary.

The efficacy of ICIs is biologically governed by cancer–immune system interactions in the tumour immune microenvironment (TIME) [5,6]. Important within this microenvironment are the CD8 glycoprotein-expressing (CD8^+^) tumour-infiltrating lymphocytes (TILs), which play a key role in destroying cancer cells [7]. Given this role after the administration of ICIs (Figure 1), it follows that the presence of CD8^+^ TILs may predict treatment response. Studies have shown that lesions with higher numbers of CD8^+^ TILs tend to be more sensitive to ICIs [8,9], and there is mounting evidence supporting the clinical utility of CD8^+^ TILs as a biomarker in various cancers [10,11,12,13,14,15]. Circulating CD8^+^ T cells in peripheral blood have been used to predict ICI response [16], but it remains elusive whether systemic measurements accurately reflect the local tumour microenvironment [9]. The gold standard for CD8^+^ TIL quantification is the assessment of biopsy specimens via immunohistochemistry [17]; this approach, however, is limited by its invasiveness, unrepeatability, and inability to reflect intra- and intertumoural heterogeneity. These shortcomings demonstrate a need for innovation in the form of dynamic and less invasive biomarkers.

The emerging field of radiomics centres on the premise that quantitative features observed in medical images may have biological underpinnings. Tumour regions with distinct radiomic phenotypes have been postulated to be representative of different tumour microenvironments [18,19] and various TIL gene expression patterns [20]. Tumours manifesting shape and texture irregularities have also been reported to show better ICI response [21]. The identification of a constellation of imaging features (“radiomic signature”) linked to CD8^+^ TILs is desirable: it could create a novel way of evaluating treatment outcomes in a manner that is non-invasive and complementary to routine patient management, as the collection of imaging data is undertaken as part of existing standards of care. Furthermore, radiomics-based biomarkers can be computed repeatedly, and permit the 3D evaluation of the entire investigated tumour lesion. While promising, it is also worth highlighting that radiomics research is characterised by substantial methodological heterogeneity [22]. Many have called for the adoption of rigorous, transparent, and standardised workflows to ensure the reproducibility of radiomic signatures [23,24,25,26,27].

Based on the ever-increasing advances in computational power and the recent clinical adoption of ICIs [28], we hypothesised that several radiomics studies involving CD8^+^ TILs have likely materialised. Moreover, little is known about the methodological robustness of these studies. In this systematic review, our objectives were three-fold: (i) to provide an overview of the general characteristics of radiomics studies involving CD8^+^ TILs, (ii) to collate radiomic signatures associated with CD8^+^ TILs reported so far, and (iii) to evaluate the methodological quality of these studies. We also discuss avenues for future investigation. The scope of this review includes radiomics studies from three diagnostic imaging modalities: computed tomography (CT), magnetic resonance imaging (MRI), and positron emission tomography (PET). 

## 2. Materials and Methods

This review was conducted in adherence to Preferred Reporting Items for Systematic Reviews and Meta-Analyses (PRISMA) 2020 guidelines [29] (Appendix A). The protocol for this review is registered with PROSPERO (CRD42021284332).

### 2.1. Literature Search

We developed our search strategy following a preliminary literature review. Identified common terms were truncated or expanded, where necessary, to account for derivational affixes and abbreviations; these were then organised into a Boolean search (Figure 2). Peer-reviewed journal articles were searched from inception until 31 December 2021, and collected from three electronic databases: Ovid MEDLINE, Embase, and Web of Science. We also included articles beyond our search that were retrieved manually, i.e., articles cited within examined literature, referred to us by experts, or encountered during previous reading.

### 2.2. Literature Selection

The shortlisted articles were initially evaluated for duplicates; the remaining were screened against eligibility criteria designed around the PICOS (Population, Intervention, Comparison, Outcomes, and Study) framework (Table 1). In summary, we sought to include only primary sources that investigated both imaging/radiomics features and CD8 data on human tumour lesions. To this end, articles were evaluated systematically at the title-, abstract-, and full-text levels, and reasons for exclusions were documented. To minimise subjectivity, screens were completed independently by two reviewers (S.R. and D.H.), and discrepancies in relevance assessment were resolved via consensus or adjudicated by a third independent reviewer (K.B.).

### 2.3. Literature Analysis

To address review objectives (i) and (ii), we carried out data extraction on all selected records. Relevant study parameters (e.g., disease, cohort size, and imaging modality) and reported radiomic signatures were tabulated (Appendix A). Extraction forms were first piloted on three studies chosen at random, then completed for all records by one reviewer (S.R.) and validated by a second reviewer (K.B.) to ensure a level of accuracy. Qualitative summaries of the general study characteristics were presented. We found no a priori reason to believe CD8^+^ TIL-associated radiomic signatures for one disease site will be similar for others. Thus, in meeting objective (ii), publications were first grouped according to the organ (or organ system) where the investigated cancer originated. Within groups, relationships between signatures were explored where possible. 

To address review objective (iii), articles were assessed using the Quality Assessment of Diagnostic Accuracy Studies version 2 (QUADAS-2) [30] and Radiomics Quality Score (RQS) [25] tools. In accordance with QUADAS-2 scoring design, we evaluated potential risks of bias and/or the applicability of the included studies in four domains: patient selection, index test, reference standard, and flow and timing. On the other hand, the RQS assigns an overall score that is reflective of the methodological quality of a radiomics study. This score, which can range from −8 to 36, is derived from the summation of ratings for 16 dimensions. Each dimension represents a key component in the radiomics pipeline (e.g., imaging protocol quality). Both tools have their own strengths: the QUADAS-2 tool is more well-established in systematic reviews, while the RQS is more specific to radiomics. We used both tools to ensure a spectrum.

Quality assessments were completed by two reviewers (S.R. and D.H.) and the inter-rater agreement was assessed. For every QUADAS-2 domain, the percentage of absolute agreement of ratings between the two reviewers was measured. The inter-rater agreement of the overall RQS was calculated using an intraclass correlation coefficient (ICC) estimate based on a mean-rating, absolute agreement, two-way, mixed-effects model [31]. For each of the RQS dimensions, the agreement of ratings between reviewers was assessed by means of a linearly weighted Gwet’s *AC*_2_ statistic for ordinal data [32]. This inter-rater reliability measure was chosen to circumvent known paradoxical behaviours associated with the more commonly used Kappa statistics [33,34]. Results were then arbitrated between reviewers. Analysis was performed using the “*irr*” and “*irrCAC*” packages in R software (R Foundation for Statistical Computing, Vienna, Austria, https://www.R-project.org/) (accessed on 16 June 2022) (v4.0.5).

## 3. Findings

### 3.1. Literature Selection 

The study selection process is presented in Figure 3. Our search yielded 1044 articles, of which 2 were manually retrieved from other sources; 297 articles were initially removed on the basis of duplication, and 611 were excluded upon evaluation at the title level, 70 at the abstract level, and 39 at the full-text level, culminating into a final 27 publications for analysis in this review. The results from our literature search and individual screens are available in Appendix A. The majority of the selected studies were published in the last two years, confirming our initial hypothesis that radiomics involving CD8^+^ TILs is a new research avenue.

### 3.2. General Characteristics of Included Studies

The main characteristics of the included articles are presented in Table 2 and a typical study workflow has been illustrated in Figure 4. Technical radiomics terminology used in this review has been defined in Appendix A. 

Studies have so far focused on cancers of the lung (7/27), hepatobiliary system (6/27), brain (4/27), gastrointestinal tract (3/27), and head and neck (2/27). Two articles investigated multiple cancers (2/27). Single studies were found on breast cancer, melanoma, and undifferentiated pleomorphic sarcoma (UPS). The median total number of patients investigated was 105 (range: 14–1778). No studies declared the use of prospectively acquired datasets; datasets tend to be sourced locally from a single institution (18/27) and/or downloaded from public repositories (such as The Cancer Genome Atlas/The Cancer Imaging Archive [35]) (6/27).

To assess the CD8 marker on tumour samples, immunohistochemistry (IHC) was performed more commonly than RNA sequencing (RNA-seq) (17/27 vs. 5/27). CD8^+^ TILs were enumerated via fluorescence-activated cell sorting (FACS) in two papers [36,37]. Three articles estimated CD8^+^ TILs via cell type quantification tools on bulk tumour transcriptome data [38,39,40]. One article used a chemokine gene expression signature as a surrogate marker for CD8^+^ TILs [41]. We highlight six studies that did not investigate CD8^+^ cells as an exclusive biological correlate; these articles instead interlaced CD8^+^ TILs with other immune variables to perform joint analyses (e.g., with CD3 [42,43,44], CD4 [36], PD-L1 [45,46] markers).

Studies analysing MRI and CT images were represented almost equally (11/27 vs. 10/27). Contrast-enhanced images predominated both modalities (9/11 and 8/10). The remaining six studies analysed PET images with fluorine-18-labelled fluorodeoxyglucose ([^18^F]-FDG) tracers ([^18^F]-FDG-PET) [45,46,47,48,49,50]. To extract intratumoural imaging features, lesion boundaries were annotated mainly via manual (15/27) or semi-automatic (10/27) means. In addition, we located three studies where the tumour periphery was considered a separate entity and the extraction of peritumoural features was carried out [43,44,51]. The software platform for radiomic feature computations was inconsistent, with at least eight different packages identified. PyRadiomics was the most commonly used software platform (8/27).

There was remarkable diversity in the types of imaging features extracted. We distinguished them into three classes. The first of these classes represents conventional radiomic features, where studies extracted the following feature families: size- or shape-based (21/27), first-order (25/27), second-order (22/27), and higher-order (9/27). Broadly, second-order families have been calculated from various matrices that describe how homogeneous or heterogeneous an image is. These matrices include the grey-level co-occurrence matrix (GLCM) (22/27), grey-level run-length matrix (GLRLM) (21/27), grey-level size zone matrix (GLSZM) (20/27), neighbouring grey-tone difference matrix (NGTDM) (16/27), grey-level dependence matrix (GLDM) (12/27), and neighbouring grey-level dependence matrix (NGLDM) (2/27). Higher-order families refer to the extraction of features from images pre-processed with various mathematical filters. 

The second feature class corresponds to semi-quantitative features, examined in eight articles. These features differ from conventional radiomics in their calculation and are more directly interpretable (e.g., total lesion glycolysis calculated from PET image intensities). The final feature class describes semantic features, studied in nine articles. These are features perceived qualitatively by radiologists (e.g., tumour location). Brief descriptions of each feature class, family, texture, and their hierarchy are provided in Appendix A. 

Radiomic signatures associated with CD8^+^ TILs were determined in one of two ways, as illustrated in Figure 4. In the predominant approach (Pipeline A), imaging features were subjected to association analysis with different CD8^+^ TIL levels, in order to single out features that are statistically different between lesions exhibiting distinct CD8^+^ TIL levels. Alternatively, radiomic signatures could also be developed by assessing the association of features with clinical variables, before being evaluated for their biological significance, i.e., link to CD8^+^ TILs (Pipeline B). Regardless of which approach was taken, most studies also employed a further feature selection step to remove features that were redundant and/or sensitive to other parameters in the radiomics workflow (23/27). Details of derived radiomics signatures have been summarised in Table 2 and discussed in the sections that follow; a more detailed list is also available in Appendix A. 

Statistical tests and threshold values were marked by heterogeneity and were highly dependent on individual study context, objectives, and end points. Study end points include using radiomic signatures to build models that could predict high or low CD8^+^ TIL levels in lesions, TIME phenotypes, survival (overall, disease-free, metastasis-free, or progression-free), and/or treatment response (Table 2). In approximately two-thirds of the papers, the predictive capacity of radiomics signatures was validated (19/27). Validation patient cohorts were mainly sourced internally, i.e., using a portion of the same patient cohort (set aside at the beginning of the study) for testing (15/19). In this case, the mean proportion of patients between training and validation was found to be 5:2. In contrast, only four articles performed external validation with datasets originating outside the researching institution [44,51,52,53].

**Table 2 cancers-14-03656-t002:** Details of included studies.

Disease		Study [Ref]	Total Cohort #	Validation	CD8 Evaluation	Joint Analysis	Imaging Modality	Radiomics Software	Features Extracted			Tumour Region	Relevant Radiomic Signatures		Modelling	End Point
									R	S	SQ		#	Features		
Lung	NSCLC	Lopci et al. [47]	55	N	IHC	N	PET ([^18^F]-FDG)	NA	N	N	Y	Intratumoural	2	SQ: SUV_mean_, SUV_max_	Cox regression	DFS
	NSCLC	Castello et al. [48]	44	N	IHC	N	PET ([^18^F]-FDG)	LIFEx	Y	N	Y	Intratumoural	5	R: First-order SQ: SUV_max_, SUV_peak_, SUV_mean_, MTV	Cox regression	DFS
	NSCLC	Mazzaschi et al. [42]	100	Y	IHC	Y (CD3, PD-1)	CT	SlicerRadiomics	Y	Y	N	Intratumoural	11	R: First-order, GLCM, GLRLM, GLDM S: Texture, effect (parenchyma reaction), margins	Cox regression	OS, DFS
	NSCLC	Mitchell et al. [49]	59	N	IHC	N	PET ([^18^F]-FDG)	NA	N	N	Y	Intratumoural	0	None significant	Cox regression	OS, DFS
	NSCLC	Zhou et al. [45]	91	N	IHC	Y (PD-L1)	PET ([^18^F]-FDG)	NA	N	N	Y	Intratumoural	5	SQ: SUV_max_, SUV_mean_, TLG	Logistic regression	Tumour immuno- phenotype
	NSCLC	Min et al. [37]	97	Y	FACS	N	CT	PyRadiomics	Y	Y	N	Intratumoural	4	R: GLCM, GLDM S: Boundary type, lymphatic metastasis	Neural network-based	High/low CD8 levels
	NSCLC	Zhou et al. [46]	103	Y	IHC	Y (PD-L1)	PET/CT ([^18^F]-FDG)	LIFEx	Y	N	Y	Intratumoural	1	R: NGLDM	Logistic regression	Tumour immuno-phenotype
Hepatobiliary	PDAC	Li et al. [54]	184	Y	IHC	N	CE-CT	PyRadiomics	Y	Y	N	Intratumoural	11	R: First-order, GLSZM S: Tumour size	Logistic regression, XGBoost	High/low CD8 levels
	PDAC	Bian et al. [55]	156	Y	IHC	N	MRI (T1W, T2W, post-contrast [AP PPP, PVP])	PyRadiomics	Y	Y	N	Intratumoural	14	R: First-order, GLCM, GLRLM, GLSZM, NGTDM S: Lesion location, tumour size	Linear regression, XGBoost	High/low CD8 levels
	PDAC	Bian et al. [56]	144	Y	IHC	N	MRI (T1W, T2W)	PyRadiomics	Y	Y	N	Intratumoural	13	R: First-order, GLCM, GLRLM, GLSZM	LDA classifier	High/low CD8 levels
	HCC	Chen et al. [43]	207	Y	IHC	Y (CD3)	MRI (CE)	Analysis Kit (GE Healthcare)	Y	N	N	Intratumoural, peritumoural, combined	70	R: Shape, GLCM, GLRLM, GLSZM	Extra-Trees, logistic regression	Immunoscore prediction
	HCC	Liao et al. [57]	142	Y	IHC	N	CE-CT	Analysis Kit (GE Healthcare)	Y	N	N	Intratumoural	7	R: GLCM, GLRLM	Elastic-net	OS, DFS
	ICC	Zhang et al. [58]	78	N	IHC	N	MRI (T1W, T2W, post-contrast [AP, PVP], DW)	PyRadiomics	Y	N	N	Intratumoural	4	R: Shape, first-order, GLSZM	Logistic regression, Cox regression	Tumour immuno-phenotype, OS
Brain	LGG	Zhang et al. [38]	107	Y	TIMER	N	MRI (T1W, T1CE, T2W, T2-FLAIR)	CaPTK	Y	N	Y	Multiple subregions	3	R: Shape, GLRLM	Cox regression	OS
	GBM	Hsu et al. [59]	116	Y	RNA-seq	N	MRI (T1CE, DW)	ND	Y	N	N	Intratumoural	15	R: First-order, GLRLM	Logistic regression	High/low CD8 levels
	HGG	Kim et al. [36]	51	N	FACS	Y (CD4)	MRI (T1W, T1CE, T2W, T2-FLAIR, DW, DSC)	PyRadiomics	Y	N	N	Intratumoural	5	R: GLCM, GLRLM, GLSZM, GLDM	sPLS-DA	OS
	Glioma	Chaddad et al. [40]	151	Y	CIBERSORT	N	MRI (T1W, T1CE, FLAIR, T2W)	MATLAB	Y	Y	N	Intratumoural	3	R: GLSZM	Neural network-based	High/low CD8 levels
Gastrointestinal	Gastric cancer	Jiang et al. [44]	1778	Y	IHC	Y (CD3, CD45RO, CD66b)	CE-CT	MATLAB	Y	N	N	Intratumoural, peritumoural	13	R: Shape, GLCM, GLRLM, GLSZM, NGTDM	Logistic regression, Cox regression	Immunoscore prediction, DFS, OS
	ESCC	Wen et al. [60]	220	Y	IHC	N	CE-CT	IBEX	Y	N	N	Intratumoural	8	R: First-order, GLCM, GLRLM	Logistic regression	High/low CD8 levels
	Rectal cancer	Jeon et al. [61]	113	Y	IHC	N	MRI (T2W)	MATLAB	Y	N	N	Intratumoural	6	R: First-order, GLCM, GLRLM, GLSZM	Linear regression	Chemoradiotherapy-induced changes
Head and neck	HNSCC	Katsoulakis et al. [52]	160	Y	RNA-seq	N	CE-CT	Radiomics Toolbox in CERR	Y	N	N	Intratumoural	67	R: First-order, GLCM, GLRLM, GLSZM, NGTDM, NGLDM	Random forest	High/low CD8 levels
	HNSCC	Wang et al. [41]	71	Y	Chemokine gene expression	N	CE-CT	SlicerRadiomics	Y	N	N	Intratumoural	8	R: GLCM, GLSZM, GLDM, NGTDM	Logistic regression	Tumour immuno-phenotype
Multiple	Multiple	Sun et al. [51]	491	Y	RNA-seq	N	CE-CT	LIFEx	Y	Y	N	Intratumoural, peritumoural	8	R: First-order, GLRLM S: Lesion location (adenopathy; head and neck), CT parameters (kVp)	Elastic-net	Objective response, OS
	Multiple	Ligero et al. [53]	198	Y	IHC	N	CE-CT	PyRadiomics	Y	Y	N	Intratumoural	16	R: Shape, first-order, GLCM, GLDM S: Lesion location (liver; other)	Elastic-net, Cox regression	Objective response, OS
Others	Breast cancer	Arefan et al. [39]	73	Y	MCP-Counter	N	MRI (DCE)	PyRadiomics	Y	N	Y	Intratumoural	2	R: Shape SQ: Tumour mean peak enhancement	XGBoost	High/low CD8 levels
	UPS	Toulmonde et al. [62]	14	N	IHC; RNA-seq	N	MRI (T1CE)	OleaSphere® Software	Y	N	N	Intratumoural	9	R: First-order, GLRLM	Cox regression	OS, MFS
	Melanoma	Aoude et al. [50]	52	N	RNA-seq; mIF; Histomorphometry	N	PET/CT ([^18^F]-FDG)	NA	Y	N	Y	Intratumoural	1	R: First-order	Cox regression	OS, PFS

Acronyms: AP = arterial phase; CaPTK = cancer imaging phenomics toolkit; CE = contrast-enhanced; CE-CT = contrast-enhanced CT; CERR = computational environment for radiological research; CIBERSORT = cell-type identification by estimating relative subsets of RNA transcript; DCE = dynamic contrast-enhanced; DFS = disease-free survival; DSC = dynamic susceptibility contrast-enhanced; ESCC = esophageal squamous cell carcinoma; Extra-Trees = extremely randomized tree algorithm; FACS = fluorescence-activated cell sorting; GBM = glioblastoma; HCC = hepatocellular cancer; HGG = high-grade glioma; HNSCC = head and neck squamous cell carcinoma; ICC = intrahepatic cholangiocarcinoma; IHC = immunohistochemistry; kVp = peak kilovoltage; LDA = linear discriminant analysis; LGG = lower-grade glioma; MCP-counter = microenvironment cell populations-counter; MFS = metastasis-free survival; mIF = multiplex immunofluorescence; N = no; NA = not available/applicable; ND = not declared; NSCLC = non-small cell lung cancer; OS = overall survival; PDAC = pancreatic ductal adenocarcinoma; PFS = progression-free survival; PPP = pancreatic parenchymal phase; PVP = portal venous phase; R = radiomic features; RNA-seq = RNA sequencing; S = semantic features; sPLS-DA = sparse partial least squares discriminant analysis; SQ = semi-quantitative features; SSF = spatial scaling factor; T1-FLAIR = T1-weighted fluid attenuated inversion recovery; T1CE = T1-weighted contrast-enhanced; T1W = T1-weighted; T2-FLAIR = T2-weighted fluid attenuated inversion recovery; T2W = T2-weighted; TIMER = tumour immune estimation resource; UPS = undifferentiated pleomorphic sarcoma; XGBoost = binary logistic extreme gradient boosting framework; Y = yes.

### 3.3. Radiomic Signatures

#### 3.3.1. Lung Cancers

In lung cancer, all seven studies examined non-small cell lung carcinoma (NSCLC). The earliest two papers hypothesised [^18^F]-FDG-PET features could reveal a link between metabolic activity and the presence of CD8^+^ TILs in neoplastic tissue [47,48]. The authors demonstrated that semi-quantitative [^18^F]-FDG-PET features (maximum and mean standardised uptake values) and a first-order feature (entropy) were associated with CD8^+^ TIL expression. Later studies, however, showed indefinite or weaker correlations [45,46,49]. The association between [^18^F]-FDG-PET features and CD8^+^ TILs, therefore, remains poorly defined.

In CT, a study reported that CD8^+^ TILs were significantly correlated with measures of texture heterogeneity (NGLDM contrast) [46]. In another paper, TIL levels co-expressing the CD8 and CD103 marker could be predicted by measures of homogeneity and high grey-level values (validation AUC = 0.753, 95% CI: NA) [37]. Higher-order CT radiomic features, generally describing the distribution of grey-level voxels (grey-level range, high emphasis, long run lengths), were significantly correlated with TIME parameters estimated from relative levels of CD8^+^, CD3^+^, and PD-1^+^ TILs [42]. Across the studies, no features were reproducible.

#### 3.3.2. Hepatobiliary Cancers

A series of pancreatic ductal adenocarcinoma (PDAC) studies published by Bian, Y. and colleagues analysed images acquired via contrast-enhanced CT [54], contrast-enhanced MRI [55], and non-contrast MRI [56]. All three studies were aimed at predicting lesion CD8^+^ TIL levels. Validation AUCs were similar at 0.705–0.790. Notably, a higher-order feature (wavelet-filtered first-order median) was associated with CD8^+^ TILs in both T2-weighted and contrast-enhanced T1-weighted MR images. This remains the only reproducible feature that we could identify in the entirety of this review.

Two teams of investigators examined hepatocellular carcinoma (HCC) in different imaging contexts (MRI vs. CT) [43,57]. In both studies, GLCM and GLRLM textures appear to be important constituents of derived radiomic signatures. High grey-level values on CT (short and long run lengths) were good determinants of CD8^+^ TIL levels (validation AUC = 0.705, 95% CI 0.547–0.863) [57]. Measures of fine textures (short and irregular run lengths) with deep texture grooves (contrast/inertia) on MRI could predict the density of CD8^+^ and CD3^+^ TILs in the tumour centre and invasive margins [43]. Notably, predictive performance improved when peritumoural features were added into models (validation AUC = 0.899 vs. 0.640).

In the context of intrahepatic cholangiocarcinoma (ICC), tumour flatness and higher-order radiomic families (variability of size zone volumes and first-order medians) in preoperative MR images could predict CD8^+^ TILs [58]. The AUC was 0.919 but a validation study was not performed. 

#### 3.3.3. Brain Cancers

The standard of care modality for the radiographic evaluation of neurologic diseases is MRI. Accordingly, all four studies on brain cancer analysed MR images. In two of these papers, high-grade gliomas (HGGs) were investigated in apparent diffusion coefficient (ADC) maps obtained from diffusion-weighted MRI. In this context, radiomic signatures between studies were dissimilar: first- and second-order GLRLM features (short runs) were good predictors of different cytotoxic TIL levels in glioblastoma (AUC = 0.710; 95% CI: NA) [59], while second-order GLSZM features (variance of grey-levels within size zone volumes) could determine CD8+ TIL-dominant HGG lesions [36]. 

Two HGG studies interrogated contrast-enhanced MR images and produced radiomic signatures that were also discordant: only first-order features were correlated with CD8^+^ TILs in one study [59] and second-order features in the other [36]. When considering only lower-grade gliomas (LGGs), second-order features (GLRLM long grey-level runs) and volume-based features have been reported to predict CD8^+^ TILs [38]. However, when LGGs and HGGs were pooled together, no features from contrast-enhanced MRI were significantly different between low and high groups of CD8^+^ TILs (*p* > 0.05) [40]. Here, authors instead demonstrated that fine textures and large size zones with high grey-levels were significant predictors of CD8^+^ TILs in non-contrast MR images. 

#### 3.3.4. Gastrointestinal Cancers

The study focusing on esophageal squamous cell carcinoma (ESCC) revealed that first-, second-, and higher-order features describing grey-level distribution (e.g., interquartile range, entropy, cluster prominence) and fine textures (short runs) could predict CD8^+^ TILs (validation AUC = 0.728, 95% CI: 0.562−0.894) [60]. 

Jiang et al. analysed the utility of intratumoural and peritumoural radiomic features to predict tumour and/or invasive margin levels of CD8^+^, CD3^+^, CD45RO^+^, and CD66b^+^ immune cells in gastric cancer (validation AUC = 0.766, 95% CI: 0.669–0.863) [44]. The radiomic signature was mainly composed of heterogeneity measures (from second- and higher-order radiomic families).

Distinct from all the other articles we reviewed, Jeon et al. performed a delta-radiomics [25] MRI study on rectal cancer to predict chemoradiotherapy (CRT)-induced changes in CD8^+^ TILs (AUC = 0.824, 95% CI: 0.674–0.974) [61]. The radiomic signature was in part defined by homogeneity measures (large areas with low grey-levels and short runs with high grey-levels). The net change in radiomic feature values between pre-CRT and post-CRT datasets was correlated with a higher longitudinal fold change in CD8^+^ TIL density (*p* = 0.001).

#### 3.3.5. Head and Neck Cancers

Radiomic signatures from contrast-enhanced CT images could only moderately predict levels of CD8^+^ TILs in head and neck squamous cell carcinoma (HNSCC) [41,52]. Second-order features describing texture heterogeneity (contrast, coarseness, small grey-level dependence) had a modest validation AUC of 0.643 (95% CI: 0.340–0.946) in one study [41], while a radiomic cluster of 67 features could only classify lesion CD8^+^ TIL levels with an accuracy of 65.7% [52].

#### 3.3.6. Multiple Cancers

Both Sun et al. and Ligero et al. used contrast-enhanced CT images to predict ICI response in datasets composed of a mixture of advanced solid tumours [51,53]. Prediction performances using derived radiomic signatures were similar, achieving good validation AUCs of 0.67–0.76. Both signatures contained lesion location (semantic feature) to account for the heterogeneity in the cancer type or organ region analysed. Additionally, in both studies, tumour homogeneity measures were associated with high CD8^+^ TIL levels or responses to ICIs. However, the included parameters were calculated from different second-order matrices (GLRLM vs. GLCM and GLDM). There was little overlap in other respects; no features were reproducible between studies, while Sun et al. accounted for and retained peritumoural features. 

#### 3.3.7. Other Cancers

In the remaining evidence, a contrast-enhanced MRI study with fourteen UPS patients reported that first-order features and fine textures (abundance of short runs, especially with high grey-levels) could predict lesion CD8^+^ TIL densities (accuracy = 93%) [62]. In a [^18^F]-FDG-PET/CT study on melanoma, only a CT first-order feature (mean value of positive pixels) could significantly identify lesion groups with distinct CD8^+^ TIL expressions (*p* = 0.017) [50]. The last study, by Arefan et al., identified that semi-quantitative features (tumour volume, mean peak enhancement of the tumour) from dynamic contrast-enhanced MRI could only moderately predict CD8^+^ TILs (validation AUC = 0.62, 95% CI: NA) [39].

### 3.4. Study Quality of Included Articles

The results from our arbitrated QUADAS-2 assessments are summarised in Figure 5A. Most papers had low risk of bias in the index test and reference standard domains due to the adequate reporting of radiomics procedures and how the CD8 marker was interrogated. That said, the risk of bias for patient selection was high, which we attribute to the intrinsic selection biases of retrospectively acquired data in the included publications. Similarly, applicability concerns for the radiomics signatures were mainly high or unclear due to the absence of a validation step or the reliance on internal validation cohorts, respectively. In the patient flow and timing domain, the majority of studies failed to report the temporal delay between imaging and pathology. 

Studies reached a mean ± standard deviation RQS of 11.81 ± 6.69 and a percentage RQS of 33.3 ± 17.5% (scores of −8 to 0 were treated as 0% and 36 treated as 100%). The RQS ranged from −2 to 22 (or 0% to 61.1%). The distribution of total scores is reflected in Figure 5B. Average ratings for each item of the RQS can be seen in Figure 5C. In summary, given that the CD8 assessment formed a criteria for literature selection, the vast majority of studies performed well in the following dimensions: multivariable analysis with non-radiomic features (27/27); discussing the potential clinical utility of findings (26/27); comparison to other or gold standard approaches (23/27), such as by assessing the added value of radiomics in clinical data-only models; performing cut-off analyses (23/27), such as by dichotomising samples into high or low CD8^+^ TIL groups based on a measured median; and discussing detected biological correlates (20/27). Most studies performed a feature selection or robustness step in the analysis (23/27). In contrast, less than half of the studies documented a comprehensive imaging protocol (11/27) (e.g., unknown voxel sizes, vendor names, tube current, and field strength). Only one study assessed the temporal variability of features by means of scanning at multiple time points [45]. Finally, we observed no phantom calibration, cost-effectiveness analysis, or provision of complete open access data (e.g., scripts, volumes of interest (VOIs), and images) in all publications reviewed.

Full quality assessments from each reviewer are available in Appendix A. The absolute agreement between reviewers was above 70% for most of the QUADAS-2 domains (Appendix A). Poor rating agreement was only seen for the risk of bias in flow and timing (37.0%) and applicability of the index test (40.7%). The ICC for the total RQS score was 0.969 (95% CI: 0.932–0.986), reflecting high agreement between reviewers. The inter-rater agreement for the dimensions of the RQS was generally good, with AC_2_ values of above 0.7 for 13 out of the 16 domains (Appendix A).

## 4. Discussion

CD8^+^ TILs are an important biomarker of ICI response. However, their assessment necessitates tumour biopsies, which are invasive and prone to sampling bias. Radiomics promises to overcome these challenges. The basic assumption of radiomics is that medical imaging features that are otherwise invisible to the naked eye could reveal underlying tumour biology. Such an assumption prompted us to systematically review and analyse studies investigating radiomic signatures associated with CD8^+^ TILs. 

We found that most studies on this topic were only published in the last two years. Still, we identified a variety of investigated tumours, many of which are established indications for ICI therapy [63]. The most studied malignancy was NSCLC [37,42,45,46,47,48,49], which was as expected given that ICIs have shifted treatment paradigms for these cancers [64]. Other cancers eligible for ICI treatment, herein, were melanoma [50], ESCC [60], gastric [44], and breast [39] cancer; yet, we have only been able to locate single papers for these to date. Some avenues currently unexplored include renal and ovarian lesions, where nomograms associated with CD8^+^ TILs have been developed in a non-radiomics context [65,66].

Our systematic review indicated radiomic signatures associated with CD8^+^ TILs are predominantly heterogeneous, despite some degree of overlap at the feature family level. Even between studies of the same cancer group, the reproducibility of radiomic features was limited. The only exception to this was a higher-order radiomic feature (wavelet-filtered first-order median), which appeared to be reproducible between two PDAC studies using different MRI protocols [55,56]. However, we highlight that both studies originated from the same institution. It would therefore be desirable to explore whether this reproducibility holds with broader datasets obtained using scanners of varying manufacturers and across multiple institutions. To complicate things further, the power of radiomic signatures in predicting CD8^+^ TILs was variable, with reported validation AUCs ranging between 0.643 and 0.899. In some papers, there does not seem to be a consensus on the association of [^18^F]-FDG-PET imaging features with CD8^+^ TILs [45,46,49]. 

We believe the lack of reproducible or definitive radiomic signatures could, at least in part, be explained by insufficiently developed and heterogeneous study methodologies. Using RQS scoring criteria, our quality assessments of included studies indicate that the study methodologies were overall less than desirable (mean RQS = 33.3%). This builds upon findings of other published radiomics systematic reviews utilising the RQS tool, wherein it was determined that radiomics research has not yet matured technically [67,68,69,70]. However, we emphasise that the low RQS scores do not necessarily devalue the impact of the reviewed articles, and merely indicate a need for more methodologically rigorous research in the future. In our review, the major reasons for the observed low scores were the use of retrospectively acquired data and the lack of results validated on external datasets. We believe these may have introduced selection biases, which was also reflected in our QUADAS-2 appraisals. To mitigate this bias and improve the generalisability of radiomic signatures, future researchers should ideally focus on validating results in large-sample, multi-institutional, and prospective settings.

Our review has revealed a remarkable diversity as regards the methods used in the included studies. This could also be evidenced by the heterogeneity of RQS scores, which range from 0% to 61.1%. Indeed, radiomic pipelines should be more harmonised to allow the better comparability of radiomic signatures between studies, and for us to reach more meaningful conclusions. A prime example illustrating the methodological variation between studies is the wide selection of software platforms used for radiomic computations. Recently, it has been reported that different software platforms could yield non-identical radiomic feature calculations and, thus, variable radiomic signatures [71,72,73]. Fortunately, community-wide efforts are ongoing to standardise feature calculations through the Imaging Biomarker Standardisation Initiative (IBSI) [26], and prospective investigators should therefore aim to use IBSI-compliant software. 

Deriving reproducible radiomic signatures is recognised as one of the major challenges to the translation of radiomics into the clinic [74,75]. Image acquisition and post-processing standardisation strategies to improve the reproducibility and clinical translatability of radiomic features have been discussed extensively in review articles by Park et al. and Vallières et al. [76,77]. For instance, the batch effect correction method “ComBat” has recently been demonstrated to substantially reduce inter-scanner biases [78,79], thus allowing for the large-scale harmonisation and pooling of inhomogeneous cohorts [80]. Furthermore, efforts to simplify radiomics workflows, in particular by automating lesion segmentation and feature processing steps via deep learning, promise to minimise the effect of variable clinical practices on radiomic signatures [81,82]. Ultimately, we again highlight the importance of high-powered prospective studies, with the expectation that a large enough sample size could overcome the inherent heterogeneities in clinical imaging [83]. Therefore, wide-reaching collaborations in the form of multi-institutional and/or multi-national consortia that offer federated imaging platforms and curated data (e.g., the EuCanImage project [84] and UK National Cancer Imaging Translational Accelerator network [85]) are also critical to facilitate the translation of imaging biomarkers into clinical practice [83].

In the following paragraphs, we describe some supplemental lines of inquiry that could be addressed by future investigators, as also illustrated in Figure 6.

First, a limited number of papers analysed features from the tumour periphery. Signatures from peritumoural regions appear to be important, or have made a positive impact on predictive modelling in the cases we reviewed [43,44,51]. Elsewhere, a growing number of studies have reported the utility of peritumoural features in their radiomics analyses [86,87,88,89]. It is also known that the response to ICI is partly dependent on the degree and localisation of CD8^+^ T-cells in the tumour margins [8,90,91]. All of this evidence, taken together, creates a strong rationale for prospective investigators to carry out radiomic interrogations of the peritumoural regions. 

Second, only a single report has so far compared features between pre- and post-treatment scans [61]. If data from multiple time points are available, prospective investigators could explore how radiomic features evolve with an underlying marked change in lesion CD8^+^ TIL density. 

Third, in the reviewed studies, associations between radiomic features and CD8^+^ TILs often completely disregard intratumoural *spatial* heterogeneity. A standard radiomics extraction pipeline computes a single average value for a given feature type, and relies on the assumption that this value is representative of the phenotype of the entire investigated lesion. Similarly, histological assessments of CD8^+^ TILs are carried out on tissue samples, which ultimately capture only a snippet of lesion biology. Fortunately, ways to address the loss of information on intratumoural spatial heterogeneity are emerging. New radiomic extraction methods can generate spatial radiomic maps that identify tumour regions presenting distinct or similar radiomic features (radiomic “habitats”) [83,92,93,94]. Meanwhile, other studies have investigated the possibility of aligning in vivo images and ex vivo samples to biologically validate radiomic signatures in space [95,96,97]. 

Novel molecular imaging tools with CD8-targeted PET imaging agents are currently being investigated [98], and may permit the non-invasive and specific imaging of CD8^+^ cells. These tools allow the spatiotemporal characterisation of CD8^+^ cell-rich tumour tissues vis-à-vis voxels in the PET images. These images could be used to study relationships between radiomic habitats and CD8^+^ TIL expression patterns in vivo. Such a relationship then opens up the possibility of developing surrogate radiomic markers of CD8^+^ TIL distribution using more conventional imaging methods. This could be especially useful for centres where next-generation imaging tools are not widely available.

Our review carries some limitations. First, the broad utility of CD8 as a marker across cancer types has led to heterogeneity in the included articles. This has ultimately precluded us from pooling data and completing a formal meta-analysis. Second, the CD8 marker is widely reputed to be a hallmark of *cytotoxic* T lymphocytes; this was assumed to be true in this present review, and by the authors of many of the included studies. However, CD8 can be expressed in other cell populations (e.g., natural killer cells [99]), and subsets of mature CD8^+^ TILs that do not exhibit the cytotoxic function also exist (e.g., regulatory or suppressor T cells [100,101,102]). How these populations influence the generalisability of findings remains to be elucidated. Third, because of the relatively small number of reviewed papers, deeper or more meaningful comparisons could not be drawn in every instance. Finally, we chose not to exclude studies with low-quality scores from our review given the limited number of papers, the emerging nature of this field, and in the interest of completeness. Thus, not all results from the publications we reviewed are free from uncertainty. 

## 5. Conclusions

In conclusion, studies deriving radiomic signatures associated with CD8^+^ TILs have recently materialised for several cancers. Observations from the reviewed studies have so far indicated that radiomic features are heterogeneous, with very limited reproducibility between studies. High-level evidence, in the form of more methodologically sound and harmonised studies, is urgently needed to generate definitive radiomic signatures and to allow the translation of radiomics into clinical practice.

## Figures and Tables

**Figure 1 cancers-14-03656-f001:**
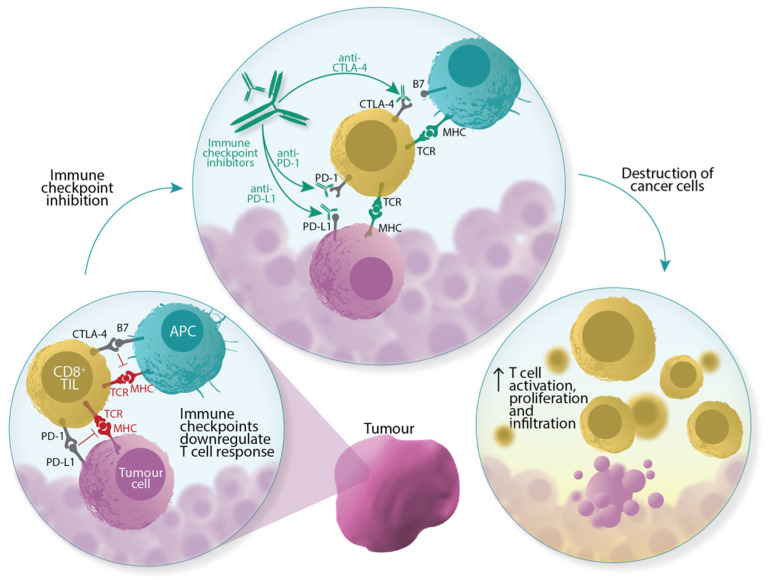
Immune checkpoint inhibitors induce tumour cell death by activating pre-existing CD8^+^ TILs. CD8^+^ TILs express T cell receptors (TCRs) that recognise antigens presented by major histocompatibility complexes (MHCs) on either tumour cells or antigen-presenting cells (APCs). TCR–antigen–MHC interactions prime and activate CD8^+^ TILs to induce apoptosis. This interaction, however, is downregulated by the activation of immune checkpoints, for example, the binding of cell surface receptor proteins PD-L1 (programmed death-ligand 1) with PD-1 (programmed death-1), and CTLA-4 (cytotoxic T lymphocyte-associated antigen-4) with B7 proteins. The blockade of these axes, via ICIs, allows CD8^+^ TILs to circumvent these inhibitory signals.

**Figure 2 cancers-14-03656-f002:**
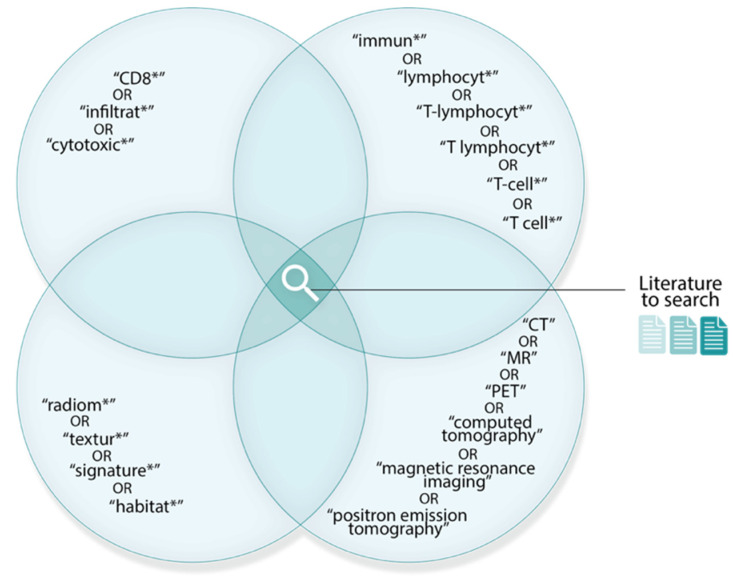
Boolean search used to retrieve relevant literature. Literature must contain at least one term from each set. Overlaps between sets indicate the Boolean AND operator. Search term truncations are denoted by an asterisk (*).

**Figure 3 cancers-14-03656-f003:**
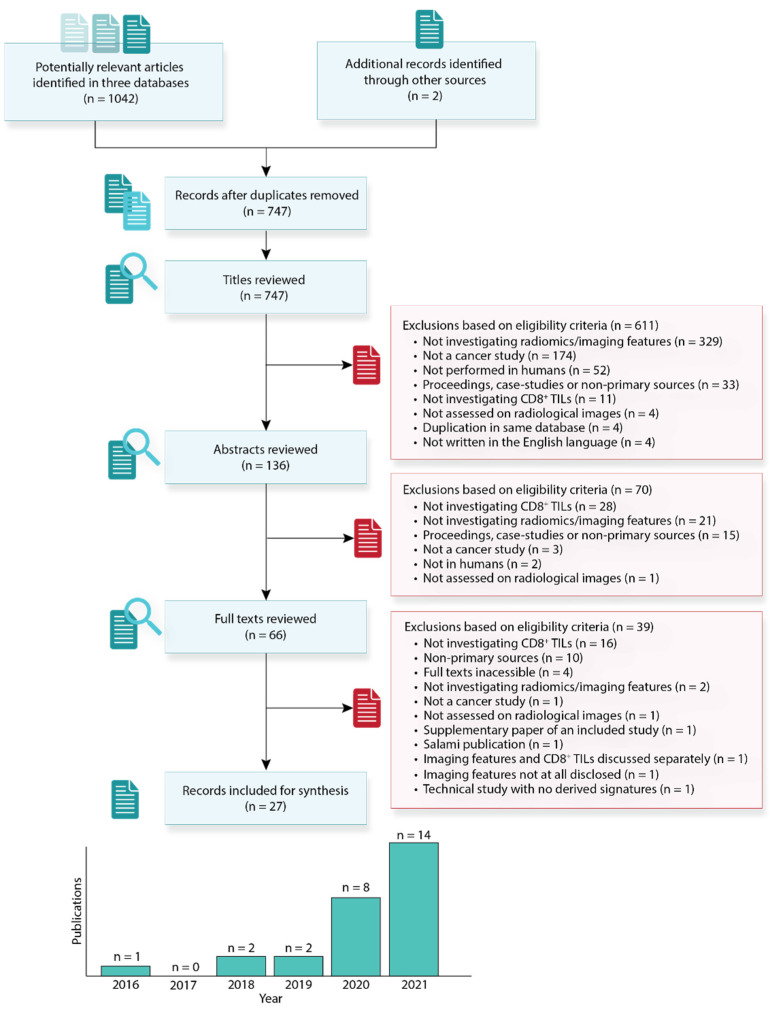
Flow diagram describing the literature selection process and the number of articles included according to the year of publication.

**Figure 4 cancers-14-03656-f004:**
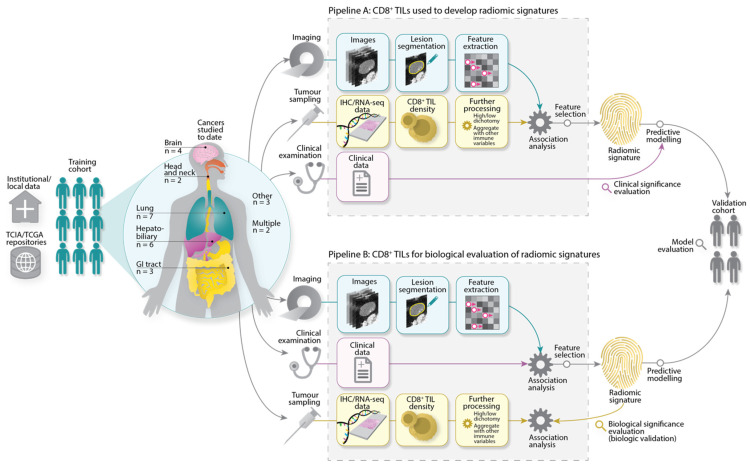
Radiomics workflow typically seen in the included studies. Imaging, biological, and clinical data were sourced from institutions and/or public repositories, before being subjected to further processing. To develop radiomic signatures, Pipeline A describes the main approach taken in the reviewed studies. Here, radiomic features were directly analysed for their association with CD8^+^ TILs. Features associated with CD8^+^ TILs were retained for radiomic signature derivation, model construction, and further evaluation. Pipeline B describes an alternate pathway where radiomic signatures were first developed by assessing the association of features with clinical variables, e.g., objective response. Signatures were then evaluated for their association with CD8^+^ TILs to explain, at least partially, the biological basis of the radiomic signatures. Acronyms: TCIA/TCGA = The Cancer Imaging Archive/The Cancer Genome Atlas, IHC = immunohistochemistry, RNA-seq = RNA sequencing.

**Figure 5 cancers-14-03656-f005:**
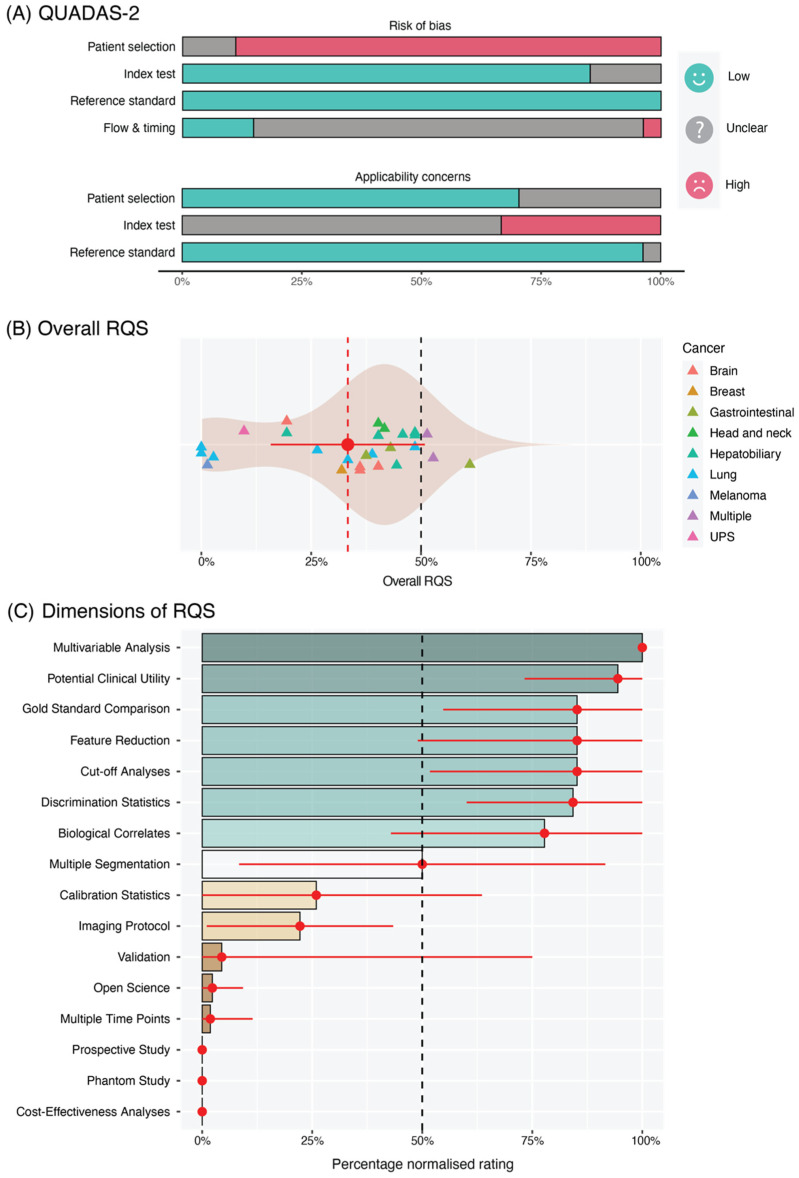
(**A**) Summary of QUADAS-2 risk of bias and applicability concern assessments after arbitration between reviewers. (**B**) Violin plot showing the distribution of the overall RQS scores achieved by reviewed studies. (**C**) Average ratings for each dimension of the RQS, normalised to percentages (0% = minimum possible positive score; 100% = maximum possible positive score).

**Figure 6 cancers-14-03656-f006:**
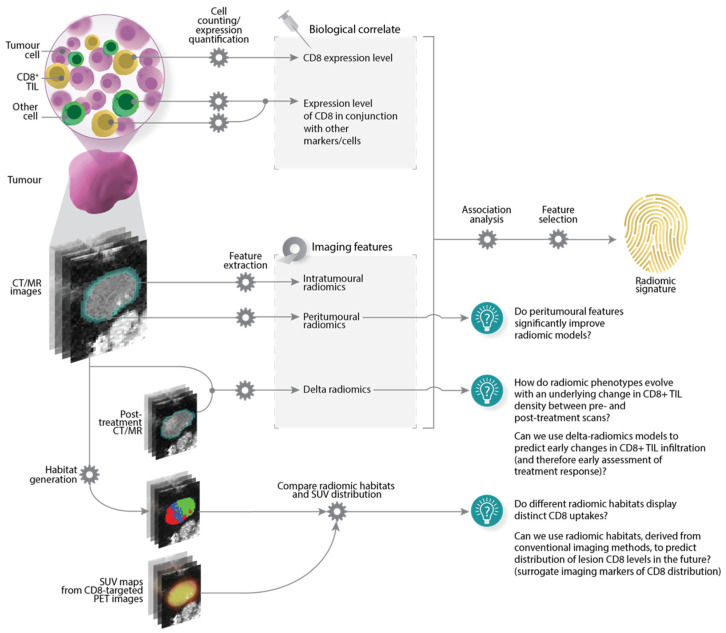
Some lines of inquiry for prospective investigators. Acronyms: SUV = standardised uptake value.

**Table 1 cancers-14-03656-t001:** Study eligibility criteria.

	Inclusion Criteria	Exclusion Criteria
**Participant(s)**	Human participants Cancer cohort	Non-human models Non-cancer studies
**Intervention(s)**	Imaging features investigated (conventional radiomics, semi-quantitative, or semantic features) Performed on radiological images (CT, MRI, and PET)	Studies not focusing on imaging features or radiomics Not performed on radiological datasets
**Comparator(s)**	CD8 marker interrogated in isolation or in combination with other markers CD8 expression measured, at least, within the tumour (e.g., via immunohistochemistry)	CD8 marker not explicitly interrogated CD8 expression not assessed within the tumour
**Outcome(s)**	Potential association of one or more imaging features (radiomic signatures) with CD8+ TILs Correlation, discrimination, or performance statistics reported (e.g., area under the curve (AUC) values) Clinically-measurable end point (e.g., survival or ICI response)	Technical studies not focused on deriving radiomic signatures
**Study design**	Primary sources	Review articles and other secondary sources Conference abstracts and proceedings Inaccessible studies Articles not written in the English language

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
