# Peer review of "Radiomic Signatures Associated with CD8+ Tumour-Infiltrating Lymphocytes: A Systematic Review and Quality Assessment Study"

_cancers, 2022, doi:10.3390/cancers14153656_

Round 1
Reviewer 1 Report
The authors propose an interesting and well documented, though not free from limitations, review of the use of radiomics for the evaluation of signatures associated with the expression of CD8 + TILs. The topic is certainly of interest and is treated with a well-set methodological approach. The authors also present in the final part of the discussion an interesting description of what could be evaluated and evaluated in future works, following their opinion. The paper is well done and well written but from the reviewer's perspective, a serious and in-depth discussion on the clinical perspective of these studies is missing. As the authors noted, in almost all tumor types, the different radiomics approaches, regardless of the quality of the studies, lack reproducibility in the obtained and documented signatures. This makes a true clinical translation of the radiomics approach impossible. It would be important to include a discussione of the clinical perspective trying to indicate some way that will allow to overcome the lack of reproducibility and lead to a true clinical translation of the results. Since without this, the correlation between radiomic biomarkers and any biological data remains a beautiful mathematical and statistical exercise without any impact on the patient's diagnostic and therapeutic workup.
Reviewer 2 Report
Ramlee and col. submitted an impressive review where they addressed the available evidence linking imaging features of tumours with levels of CD8+ TILs. The paper has an interest for broad scientific auditorium and provides interesting information. Tables and figures are very self-explicative and nicely presented. Maybe figure 4 would benefit from bigger letter size. The background and the cited literature is up-to-date and the structure is well designed. The document is excellently written. Overall, I consider the manuscript deals with an issue of interest and fits the scope of the journal. I recommend its publication in its current form.
